# Enzyme-Linked Immunosorbent Assay Using Henipavirus-Receptor EphrinB2 and Monoclonal Antibodies for Detecting Nipah and Hendra Viruses

**DOI:** 10.3390/v16050794

**Published:** 2024-05-16

**Authors:** Wenjun Zhu, Greg Smith, Bradley Pickering, Logan Banadyga, Ming Yang

**Affiliations:** 1National Centre for Foreign Animal Disease, Canadian Food Inspection Agency, 1015 Arlington Street, Winnipeg, MB R3E 3M4, Canada; wenjun.zhu@inspection.gc.ca (W.Z.); greg.smith@inspection.gc.ca (G.S.); bradley.pickering@inspection.gc.ca (B.P.); 2Department of Medical Microbiology and Infectious Diseases, University of Manitoba, Winnipeg, MB R3E 0J9, Canada; logan.banadyga@phac-aspc.gc.ca; 3Department of Veterinary Microbiology and Preventative Medicine, College of Veterinary Medicine, Iowa State University, Ames, IA 50011, USA; 4National Microbiology Laboratory, Public Health Agency of Canada, 1015 Arlington Street, Winnipeg, MB R3E 3R2, Canada

**Keywords:** Nipah virus, Hendra virus, henipavirus receptor, recombinant EphrinB2, monoclonal antibody, antigen-detection ELISA

## Abstract

The Nipah virus (NiV) and the Hendra virus (HeV) are highly pathogenic zoonotic diseases that can cause fatal infections in humans and animals. Early detection is critical for the control of NiV and HeV infections. We present the development of two antigen-detection ELISAs (AgELISAs) using the henipavirus-receptor EphrinB2 and monoclonal antibodies (mAbs) to detect NiV and HeV. The NiV AgELISA detected only NiV, whereas the NiV/HeV AgELISA detected both NiV and HeV. The diagnostic specificities of the NiV AgELISA and the NiV/HeV AgELISA were 100% and 97.8%, respectively. Both assays were specific for henipaviruses and showed no cross-reactivity with other viruses. The AgELISAs detected NiV antigen in experimental pig nasal wash samples taken at 4 days post-infection. With the combination of both AgELISAs, NiV can be differentiated from HeV. Complementing other henipavirus detection methods, these two newly developed AgELISAs can rapidly detect NiV and HeV in a large number of samples and are suitable for use in remote areas where other tests are not available.

## 1. Introduction

The Nipah virus (NiV) and Hendra virus (HeV) are zoonotic viruses of great concern, known for their high pathogenicity and mortality in animals and humans [1,2]. NiV continues to cause outbreaks in Bangladesh and India, while HeV repeatedly re-emerges in Australia. NiV and HeV infections present as severe systemic and often fatal neurologic and/or respiratory diseases in humans and several mammalian species [3,4]. NiV is of particular concern as a potential pandemic threat due to its high case fatality rate and its ability to spread from person to person [5].

NiV and HeV belong to the genus Henipavirus in the family Paramyxoviridae. Their genome encods six structural proteins: nucleoprotein (N), phosphoprotein (P), matrix protein (M), fusion protein (F), glycoprotein (G), and RNA polymerase protein (L). The two glycoproteins F and G are anchored in the envelope of the viral particle and are required for viral penetration into host cells [6]. NiV strains found in Bangladesh and India (clade I, or NiV-B) are distinguished from strains found in Malaysia and Singapore (clade II, or NiV-M) [7,8,9,10].

The susceptibility of animals and humans to NiV and HeV means that medical countermeasures are urgently needed to prevent and contain outbreaks caused by these viruses. Although an HeV vaccine has been licensed for use in horses in Australia [1], there are no vaccines or therapeutics against HeV for use in humans [5]. Likewise, there are currently no NiV vaccines licensed for use in animals or humans [11]. Therefore, early viral detection during an outbreak is critical to identify cases and limit viral spread [12]. In addition, monitoring bat populations for the presence of these viruses is also important to understand where and when viruses appear. Typical laboratory diagnostic systems available for henipavirus detection include nucleic acid-based detection, virus isolation, virus-antiserum neutralization (e.g., plaque reduction assay), and immunohistochemistry in fixed tissues. ELISAs have been widely used to detect the presence of target biomolecules. ELISAs can directly detect virus proteins in a high throughput manner with a relatively simple procedure and minimal specialized equipment.

In this study, we aimed to develop an ELISA for NiV and HeV detection using EphrinB2 as the capture ligand and monoclonal antibodies (mAbs) as the detection agent. Ephrins have been identified as functional cellular receptors for NiV and HeV. EphrinB2 has been previously reported as a capture agent for henipavirus detection [13]. However, Ephrin B2 needs to be paired with NiV- and/or HeV-specific antibodies in the ELISA to specifically detect these viruses. We have previously generated a panel of mAbs from mice immunized with BEI-inactivated NiV-M. These mAbs were further characterized, and suitable mAbs were selected for use in the henipavirus detection ELISAs described here.

## 2. Materials and Methods

### 2.1. Ethics Statement

All animal experimental procedures were performed according to Canadian Council on Animal Care guidelines. Group housing was conducted in biosafety level-4 laboratories (BSL-4) animal compartments, where the animals were provided with food and water. The animal studies reported in this manuscript complied with the Animal Research: Reporting of In Vivo Experiments (ARRIVE) guidelines.

### 2.2. Viruses

NiV-M (GenBank Accession No. AF212302), NiV-B (GenBank Accession No. AY988601.1) and HeV (GenBank Accession No. NC_001906.3) were provided by the Centers for Disease Control and Prevention in Atlanta, GA, USA. Ebola virus (EBOV; variant Kikwit) and vesicular stomatitis virus (VSV) pseudotyped with the EBOV glycoprotein (VSV-EBOV-GP) were provided by the Public Health Agency of Canada (PHAC). VSV pseudotyped with both the EBOV and NiV glycoproteins (VSV-EBOV-GP/NiV-G) was provided by Reston Microbiology Laboratory, US. Foot-and-mouth disease virus (FMDV) was provided by the National Center for Foreign Animal Disease, Canada (NCFAD). The NiV and HeV preparation procedures were described previously [14]. EBOV and VSV-EBOV-GP were prepared as previously published [15,16]. All viruses were inactivated using gamma irradiation (5 million rad) except FMDV, which was inactivated using binary ethyleneimine as previously published [17]. To denature the viruses, NiV-M, NiV-B, and HeV in culture supernatants were mixed with an equal volume of Pierce IP lysis buffer (Fisher Scientific, Waltham, MA, USA) and incubated for 30 min at room temperature.

### 2.3. Generation of Monoclonal Antibodies

The procedure for generating hybridomas was outlined previously [14]. Briefly, female Balb/c mice were inoculated subcutaneously with an equal mixture of complete and incomplete Freund’s adjuvant (Difco, BD, Oakville, ON, Canada) with purified and gamma-irradiated NiV-M (20 µg/mouse). Three identical immunizations are given at four-week intervals, with a final booster before fusion. Myeloma cells (P3 X63 Ag8.653, ATCC, Rockville, MD, USA) were fused with spleen cells from immunized mice. Hybridoma cell-culture supernatants were screened using NiV antigen. Positive clones were selected and subcloned. The two mAbs were designated F20NiV-65 and F27NiV-34 in this study.

### 2.4. Indirect ELISA

Microtitre plates (Nunc Immuno MaxiSorp, ThermoFisher, Waltham, MA, USA) were coated with recombinant NiV G (0.86 mg/mL) [18], NiV F (0.8 mg/mL, DAG-WT633, Creative Diagnostics, Shirley, NY, USA), NiV N (0.85 mg/mL, NativeAntigen Company, Kidlington, UK, Cat# REC31746), or EBOV GP (1.015 mg/mL, IBT Bioservices, Rockville, MD 20850, USA, Cat# 0511-015) in a carbonate/bicarbonate buffer (pH 9.6) overnight at 4 °C. Following overnight incubation, plates were washed five times with 0.01 M phosphate-buffered saline with 0.1% Tween 20 (PBS-T). The plates were blocked with casein-blocking buffer (Sigma-Aldrich, St. Louis, MO, USA and Burlington, MA, USA) and incubated for 60 min at 37 °C with gentle shaking. Detection mAbs (1:50) were then added, followed by HRP-conjugated anti-mouse IgG (1:2000, Jackson ImmunoResearch Laboratories, West Grove, PA, USA). Then 3,3′, 5,5′ tetramethylbenzidine dihydrochloride (TMB, Pierce Biotechnology, Inc., Rockford, IL, USA) was added. Each incubation and wash step was as described above. The reaction was stopped with 2 M sulfuric acid and an optical density at a wavelength of 450 nm (OD_450_) was read in an Emax microplate reader (Molecular Devices, San Jose, CA, USA).

### 2.5. Antigen-Detection ELISA (AgELISA)

Microtiter plates were coated overnight at 4 °C with 100 μL/well of recombinant Ephrin B2 (2 μg/mL, GenScript Inc., Piscataway, NJ, USA) in carbonate/bicarbonate buffer (pH 9.6). After blocking, inactivated or lysis buffer-treated viruses and samples (100 µL diluted in casein blocking buffer) were added and incubated. After washing, the hybridoma cell-culture supernatant (F20NiV-65, F27NiV-34) was diluted 1:50 in a blocking buffer and added to each well. Following incubation, HRP-conjugated anti-mouse IgG diluted in casein-blocking buffer (1:2000) was added and, then, TMB substrate. The reaction was stopped with 2 M sulfuric acid and OD_450_ was read. Each incubation and wash step is as described above.

### 2.6. Animal Samples

As negative controls, 55 tissues from disease-free pigs were collected from slaughterhouses in Manitoba, Canada, and six tissues from animals infected with FMDV were archived from previous animal experiments at NCFAD. A 10% suspension of each tissue was prepared as previously described [19]. A total of 80 oral and nasal swab samples were collected from a total of 40 weaned Landrace and Large White cross piglets aged 4–5 weeks (local supplier in Manitoba) prior to virus inoculation. The samples from pigs inoculated with NiV-M were archived from a previous animal experiment (unpublished). The Landrace pigs (8 weeks old) underwent at least one week of acclimatization in a large animal cubicle at NCFAD before the virus challenge. Three pigs (#5, 6, and 7) were the banked samples of NiV-M-infected animal samples from a previous study. Each pig was inoculated with a dose of 5 × 10^5^ plaque-forming units (pfu) of NiV-M in a final volume of 3 mL via the oral (1 mL) and intranasal (1 mL per nostril) routes under isofluorane gas anesthetic. Nasal wash fluid from each animal was collected at 2 and 4 days post-inoculation (dpi). All tissue samples were collected at 6 dpi.

### 2.7. qRT-PCR for Detection of NiV-M

Genotype-specific semi-quantitative real-time (q) RT-PCR was utilized to detect the RNA of NiV-M-infected samples. qRT-PCR was performed using primers and probe targeting of the viral nucleoprotein gene as described previously [20,21].

## 3. Results

### 3.1. Characterization Henipavirus-Specific mAbs

It was previously determined that F20NiV-65 recognizes recombinant NiV-G in an indirect ELISA [18]. However, the antigen recognized by F27NiV-34 has not yet been identified. To further confirm and/or identify the antigen recognized by these mAbs, we set up an indirect ELISA using the NiV recombinant structural proteins G, N, F, and EBOV GP as the coating antigens. The results showed that F20NiV-65 specifically bound NiV G, but not N, F, or EBOV GP, confirming that the binding site of F20NiV-65 is located on NiV G (Figure 1a). The F27NiV-34 mAb reacted only with the NiV F protein in a concentration-dependent manner, indicating that this antibody bound to an epitope located on NiV F (Figure 1b).

To select suitable mAbs as detection reagents, F20NiV-65 and F27NiV-34 were tested for their binding affinities and sensitivities to NiV and HeV. The results showed that the two mAbs were able to detect inactivated NiV-B with similar sensitivity using an ELISA in which EphrinB2 captured henipavirus in cell-culture supernatants (Figure 2a). Conversely, only F27NiV-34 reacted with HeV (Figure 2b). Since F20NiV-65 only recognized NiV, it was chosen as the detection mAb for the NiV-specific AgELISA. F27NiV-34 was selected as the detection reagent for the HeV/NiV AgELISA, since this antibody recognized both viruses.

### 3.2. Assay Cutoff Determination

EphrinB2 was selected as the capture agent for the AgELISAs [13]. The optimal concentration of EphrinB2 coating to achieve saturation was determined to be 2 μg/mL. A comparison of low- and high-binding plates showed that using high-binding plates captured more antigens and, therefore, increased the signal in the ELISA. The detection antibodies were titrated using a checkerboard titration, wherein both F20NiV-65 and F27NiV-34 exhibited optimal detection capabilities at a concentration of 0.6 μg/mL. To determine the correct cut-off value of the AgELISAs, a total of 135 negative samples (80 swab samples and 55 tissue suspensions) were tested. Based on negative samples, the cut-off value (defined as the mean OD_450_ plus three standard deviations) was 0.1 for both AgELISAs (Figure 3). Three negative samples showed an OD_450_ above 0.1 in the NiV/HeV AgELISA. Therefore, the calculated diagnostic specificities of the NiV AgELISA and the NiV/HeV AgELISA were 100% and 97.8%, respectively.

### 3.3. AgELISA Analytical Specificity and Sensitivity

To confirm that the AgELISAs were specific for NiV and HeV, each assay was tested with several different inactivated viruses, including NiV, HeV, FMDV, and EBOV. When F20NiV-65 was used, the AgELISA was able to detect both NiV-M and NiV-B, but it was unable to detect HeV, FMDV, or EBOV. When mAb F27NiV-34 was used, the AgELISA was able to detect NiV-M, NiV-B, and HeV, without any cross-reactivity with other viruses (Figure 4).

To evaluate whether the AgELISAs can detect denatured henipaviruses, NiV and HeV were treated with a lysis buffer and then applied to the ELISA. This treatment was conducted because previous studies have demonstrated the ability of lysis buffers to inactivate members of four different viral genera [22]. The results showed that the NiV AgELISA detected lysis buffer-treated NiV with no significant difference compared to untreated NiV, whereas the NiV/HeV AgELISA was able to detect a small portion of lysis buffer-treated NiV and HeV (Figure 4). Therefore, this suggests that F20NiV-65 binds to a linear epitope that is not destroyed by viral denaturation, whereas F27NiV-34 mostly binds to a conformational epitope that is disrupted upon lysis.

In addition, recombinant pseudotyped VSV-EBOV-GP/NiV-G viruses were tested in a NiV AgELISA, with VSV-EBOV-GP as a negative control, since only γ-irradiated virus had been tested previously. As shown in Figure 5, the OD_450_ value decreased with the increasing dilutions of VSV-EBOV-GP/NiV-G. VSV-EBOV-GP, on the other hand, could not be detected. The results confirmed that the NiV AgELISA could specifically detect NiV (Figure 5).

The limit of detection (LOD) of each AgELISA was tested using two-fold dilutions of the virus in the culture supernatant. The LOD for NiV B was 1:640, corresponding to 1.6 × 10^4^ pfu/mL, while the LOD detected by both AgELISAs was 2.6 × 10^4^ pfu/mL for NiV M. For the NiV/HeV Ag ELISA, the LOD was 1.4 × 10^5^ pfu/mL (1:320) for HeV (Table 1).

### 3.4. Detection of NiV in Experimental Animal Samples

Nasal wash and tissue samples from three pigs experimentally infected with NiV-M were collected and archived from a previous animal experiment (unpublished). All samples are listed in Table 2 and were confirmed to be NiV positive by RT-PCR. These samples were tested using the NiV- and NiV/HeV AgELISAs. Two nasal wash samples obtained at 4 dpi tested positive on both AgELISAs. The nasal wash samples obtained at 2 dpi and all tissue suspensions showed negative results on both AgELISAs (Table 2).

## 4. Discussion

In the current report, NiV-specific AgELISA and NiV/HeV-AgELISA were developed and evaluated using the henipavirus-receptor Ephrin B2 as the capture agent and two mAbs F20NiV-65 and F27NiV-34 as the detection agents. The use of EphrinB2 as a universal capture agent for henipavirus in the current ELISA is based on previous results demonstrating that EphrinB2 could capture NiV and HeV in immunoassays [13]. EphrinB2 serves as a functional receptor for henipavirus entry into host cells by interacting with the henipavirus glycoprotein [23,24,25]. Notably, the use of only EphrinB2 in immunoassays does not provide specificity for henipaviruses; therefore, there is a need to pair EphrinB2 with henipavirus-specific mAbs. Two mAbs were characterized: the mAb F20NiV-65 reacted with NiV only and F27NiV-34 recognized both NiV and HeV. The binding site of F20NiV-65 is located on NiV G, while the binding site of F27NiV-34 is located on NiV F. Henipavirus G and F are viral surface glycoproteins that are the primary antigenic targets. Therefore, F20NiV-65 and F27NiV-34 were selected as the detection reagents for the NiV AgELISA and the NiV/HeV AgELISA, respectively.

The NiV AgELISA was able to detect NiV and pseudotyped VSV-EBOV-GP/NiV-G, but not VSV-EBOV-GP, confirming that this AgELISA is NiV specific. While the NiV/HeV AgELISA detected both NiV and HeV. With the combination of these two AgELISAs, NiV can be differentiated from HeV. Neither AgELISA cross-reacted with unrelated viruses (FMDV and EBOV). Although two unrelated viruses were tested using the AgELISA to demonstrate the AgELISAs’ specificity, we were unable to detect other henipaviruses found in bats and other mammals as well as other HeV genotypes because these viruses are not available in our laboratory. If possible, these viruses will be tested in the future to further confirm that the AgELISAs are NiV- and HeV-specific. Based on 135 negative samples, the cut-off value for both AgELISAs was determined to be an OD_450_ value of ≥0.1. The diagnostic specificities of the NiVAgELISA and the NiV/HeV AgELISA were 100% and 97.8%, respectively; however, these values were determined based on a small number of negative samples. Ideally, more negative samples should be tested to obtain a more accurate cut-off and limit the number of false negatives and positives. Fine-tuned cut-off values for ELISAs should depend on the goals of the organization.

Both AgELISAs showed the same LOD. The LOD range was 1.6–2.6 × 10^4^ pfu/mL for NiV-B and M, while it was 1.4 × 10^5^ pfu/mL for HeV in the NiV/HeV AgELISA. The ELISA reported by Kaku et al. [26] used polyclonal antibodies against G and F proteins but had higher LODs for NiV-B and HeV than the current AgELISA. In general, polyclonal antibodies should capture and detect more viral antigens, although there are disadvantages to using polyclonal antibodies, including an increased chance of non-specific binding, resulting in a higher false-positive rate than monoclonal antibodies. Chiang et al. [27] reported an ELISA targeting internal N and P proteins with LODs comparable to our results. Overall, the analytical sensitivities of our AgELISAs are in a similar range to previously reported ELISAs [26,27]. Comparing analytical sensitivity between different laboratories will be difficult due to differences in viral titer assay protocols and cell types used.

The AgELISAs were shown to be able to detect viral antigens in 4 dpi nasal wash samples from NiV-M infected pigs. However, we did not capture signals in 2 dpi nasal wash samples and tissues collected from the same animals (Table 2). Although lower Ct values were detected in these samples by PCR, the detection of NiV in samples from experimentally infected pigs may be limited based on previous studies [21]. The RT-PCR Ct value of the 2 dpi nasal wash sample was higher than that of the 4 dpi sample, indicating that the viral load of the 2 dpi nasal wash sample was lower than that of the 4 dpi samples, which could explain why the AgELISAs only detected a virus in the 4 dpi samples. Possible explanations for the inability of the AgELISAs to detect the NiV antigen in PCR-positive tissues might be (1) that long-term storage and repeated freeze–thaw cycles might cause viral protein degradation, thus affecting the detection effect of AgELISA or (2) that the ELISA is not as sensitive as the RT-PCR, especially when the RT-PCR was performed with fresh samples. Although RT-PCR is currently considered the first-choice method for diagnosing NiV infections due to its high specificity and sensitivity [28], having multiple testing platforms for detecting emerging viruses would be beneficial for the rapid identification of outbreak pathogens.

Due to the limitations in obtaining negative and positive (outbreak and animal experiments) samples, we were unable to fully validate the AgELISAs. Therefore, additional testing using more negative and positive samples is required to complete the validation of the assays. These two AgELISAs are relatively simple and easy to implement in developing countries, providing a cost-effective tool in remote areas where molecular detection methods are not available, allowing for early detection of the henipavirus.

## Figures and Tables

**Figure 1 viruses-16-00794-f001:**
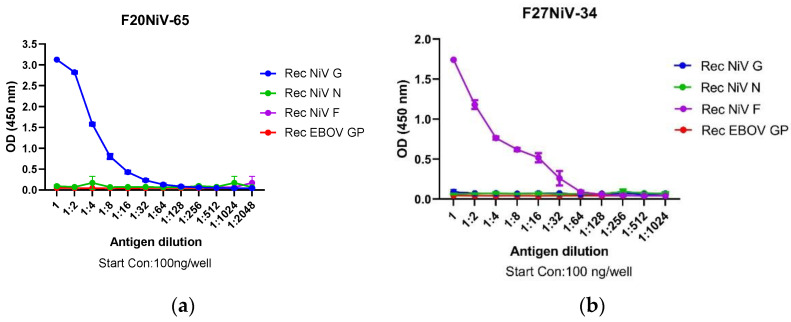
Reactivity of mAbs to recombinant NiV structural proteins in indirect ELISA. Two-fold diluted recombinant and purified NiV G, N, F, and EBOV GP were coated onto ELISA plates. After blocking, mAbs F20NiV-65 (**a**) or F27NiV-34 (**b**) were diluted at 1:50 and added to the plates. After 1 h incubation, HRP-conjugated anti-mouse IgG and a substrate TMB were added. The reaction was stopped with 100 μL of 2 M sulfuric acid and optical density at a wavelength of 450 nm (OD_450_) and was read in a microplate reader. The data shown are the mean of duplicates with error bars.

**Figure 2 viruses-16-00794-f002:**
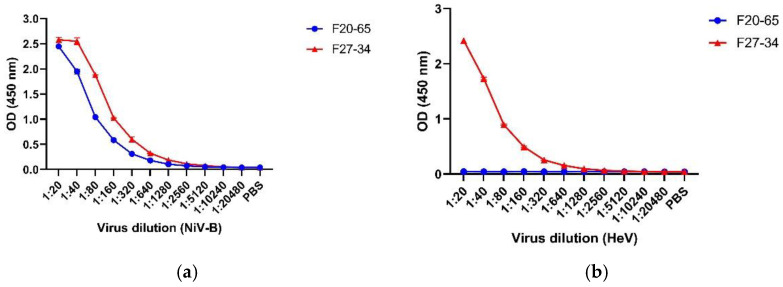
Reactivity of mAbs to NiV-B and HeV. Recombinant EphrinB2 was coated onto ELISA plates. After blocking, two-fold diluted inactivated NiV-B (**a**) or HeV (**b**) were added, and the viruses were detected using mAbs F20NiV-65 or F27NiV-34. HRP-conjugated anti-mouse IgG and a substrate TMB were added. The reaction was stopped, and the OD_450_ was read in a microplate reader. The data shown are the mean of duplicates with error bars.

**Figure 3 viruses-16-00794-f003:**
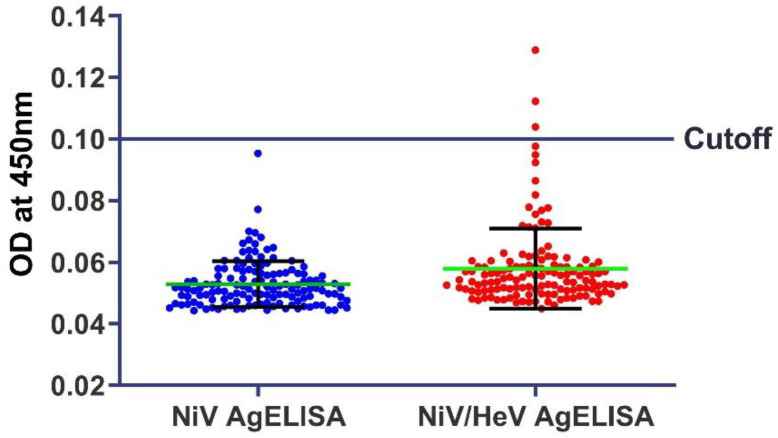
Distribution of negative samples in AgELISAs. The NiV-negative tissue suspensions (*n* = 55) and 0 dpi swab samples (*n* = 80) were tested using AgELISAs. The cut-off values of AgELISAs were determined based on the mean OD_450_ of negative samples plus three standard deviations.

**Figure 4 viruses-16-00794-f004:**
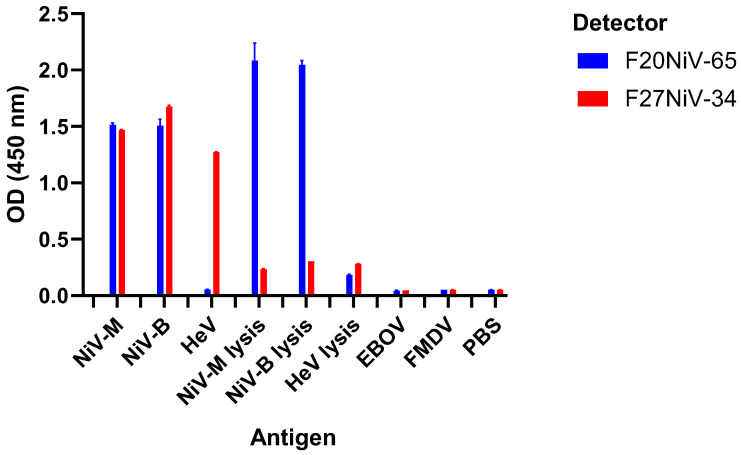
The analytical specificity of the AgELISAs. Recombinant EphrinB2 was coated onto ELISA plates. After blocking, inactivated viruses (NiV-M, NiV-B, HeV, EBOV, and FMDV) or lysis buffer-treated viruses (NiV-M, NiV-B, and HeV) were added in duplicate. The viruses were detected using mAb F20NiV-65 or F27NiV-34. HRP-conjugated anti-mouse IgG and TMB substrate were added. The reaction was stopped and OD_450_ was read in a microplate reader. The data shown are the mean of duplicates with error bars.

**Figure 5 viruses-16-00794-f005:**
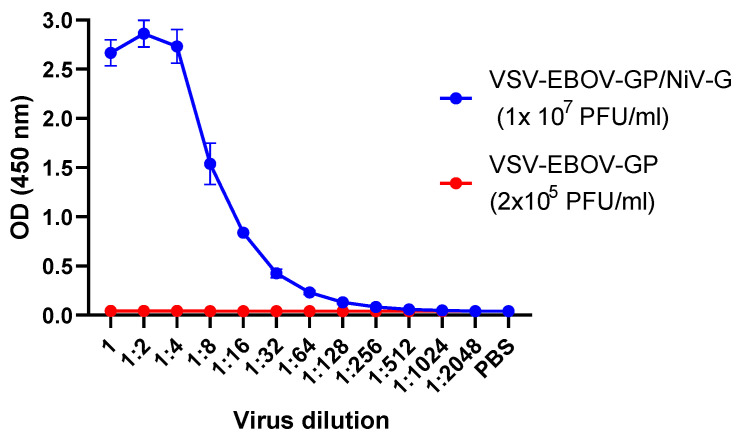
Detection of pseudotyped viruses using the NiV AgELISA. Recombinant EphrinB2 was coated onto ELISA plates. After blocking, two-fold dilutions of VSV-EBOV-GP/NiV-G or VSV-EBOV-GP were added. The bound pseudotyped viruses were detected using mAb F20NiV-65. HRP-conjugated anti-mouse IgG and TMB substrate were added. The OD_450_ was read in a microplate reader. The data shown are the mean of duplicates with error bars.

**Table 1 viruses-16-00794-t001:** Analytical sensitivity of the AgELISAs (**a**) NiV AgELISA and (**b**) NiV/HeV AgELISA.

(**a**)																		
		**NiV B**									**NiV M**							
**Virus Dilution**		**Mean of Exp #1**		**Mean of Exp #2**			**Mean of Exp 1 and 2**			**SD**	**Mean of Exp #1**		**Mean of Exp #2**			**Mean of Exp 1 and 2**		**SD**
1:20		2.51		2.28			2.39			0.13	1.73		1.98			1.90		0.20
1:40		1.84		1.68			1.76			0.08	1.55		1.05			1.22		0.26
1:80		0.99		0.91			0.95			0.04	0.57		0.55			0.56		0.06
1:160		0.56		0.46			0.51			0.05	0.31		0.28			0.29		0.03
1:320		0.28		0.25			0.27			0.02	0.17		0.13			0.14		0.02
1:640		0.15		0.13			0.14			0.01	0.09		0.07			0.07		0.01
1:1280		0.07		0.05			0.06			0.01	0.06		0.03			0.04		0.02
1:2560		0.04		0.03			0.03			0.01	0.04		0.01			0.02		0.01
(**b**)																		
	**NiV B**							**NiV M**						**HeV**				
**Virus Dilution**	**Mean of Exp #1**		**Mean of Exp #2**		**Mean of Exp 1 and 2**	**SD**		**Mean of Exp #1**	**Mean of Exp #2**		**Mean of Exp 1 and 2**	**SD**		**Mean of Exp #1**	**Mean of Exp #2**		**Mean of Exp 1 and 2**	**SD**
1:20	2.51		2.51		2.51	0.13		1.14	1.91		1.65	0.37		1.69	1.40		1.54	0.17
1:40	1.84		1.84		1.84	0.08		0.64	0.98		0.86	0.16		0.85	0.75		0.80	0.05
1:80	0.99		0.99		0.99	0.04		0.39	0.53		0.49	0.07		0.45	0.41		0.43	0.02
1:160	0.56		0.56		0.56	0.05		0.20	0.29		0.26	0.04		0.21	0.20		0.20	0.01
1:320	0.28		0.28		0.28	0.02		0.11	0.14		0.13	0.01		0.11	0.10		0.10	0.01
1:640	0.15		0.15		0.15	0.01		0.05	0.07		0.06	0.01		0.05	0.04		0.05	0.00
1:1280	0.07		0.07		0.07	0.01		0.03	0.05		0.04	0.02		0.02	0.02		0.02	0.00
1:2560	0.04		0.04		0.04	0.01		0.02	0.01		0.01	0.00		0.01	0.01		0.01	0.00

**Table 2 viruses-16-00794-t002:** Detection of Nipah virus (M) in experimentally infected pig samples using AgELISAs.

Animal #	Samples	Days Post-Inoculation	RT-qPCR (Ct)	NiV AgELISA		NiV/HeV AgELISA	
				OD_450_	Results	OD_450_	Results
P5	Nasal wash	2	28.5	N/A	N/A	N/A	N/A
P5	Nasal wash	4	26.5	N/A	N/A	N/A	N/A
P6	Nasal wash	2	27.2	0.073	-	0.0525	-
P6	Nasal wash	4	18.8	0.48	+	0.52	+
P7	Nasal wash	2	29.5	0.06	-	0.0505	-
P7	Nasal wash	4	21.5	1.14	+	1.4	+
P5	Trachea	6	20.3	0.0575	-	0.055	-
P5	Lung	6	20.6	0.0995	-	0.053	-
P5	Submandibular LN	6	16.04	0.057	-	0.0865	-
P5	Bronchial LN	6	18.8	0.122	-	0.067	-
P5	Meninges	6	19.6	0.0525	-	0.073	-
P5	Turbinate	6	18.8	0.0545	-	0.06	-
P6	Submandibular LN	6	21.4	0.0785	-	0.117	-
P6	Turbinate	6	16.8	0.0785	-	0.0575	-
P6	Lung	6	15.8	0.0535	-	0.0995	-
P6	Trachea	6	14.58	0.0505	-	0.057	-
P6	Meninges	6	17.6	0.0545	-	0.122	-

NA: not available; +: positive results; -: negative results.

## Data Availability

The data sets generated and/or analyzed in this study will be made available upon reasonable request. A material-transfer agreement may be required.

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
