# Peer review of "Enzyme-Linked Immunosorbent Assay Using Henipavirus-Receptor EphrinB2 and Monoclonal Antibodies for Detecting Nipah and Hendra Viruses"

_viruses, 2024, doi:10.3390/v16050794_

Round 1

Reviewer 1 Report

Comments and Suggestions for Authors

The authors describe an AgELISA assay to detect NiV and HeV. The importance of having an established assay and its possible uses have been described. Some of the data is similar to what has been published before by the authors. The manuscript should focus on the new information it brings.

Minor comments:

·        Line 143: This section might need a better description of the source of the samples. Three pig samples used in the study were control samples from a previously unpublished study.  Using “control animals” may be a little confusing: positive or negative control?  Maybe just say banked samples of NiV-M-infected animal samples from a previous study.

Oral and nasal swabs were collected from 40 animals prior to virus inoculation, but only three of them got infected?

·        Line 114: The concentrations of proteins used for coating ELISA plates can be added to the method section.

·        Table 1: Any IHC performed with these samples?

·        The data in Figure 1a has already been published before. What is the mAb dilution used in this assay?

·        Lines 195-201 and most of Figure 2 have been published by the authors previously.

From the previous paper:

“Given that ephrin B2 can replace polyclonal antibodies as a ligand in ELISAs to detect henipavirus, we next sought to establish a more specific antigen detection ELISA using mAbs F27NiV-34 and F20NiV-65. Since mAb F27NiV-34 reacts with both NiV and HeV, it was used to detect both henipaviruses, while mAb F20NiV-65, which reacts only with NiV, was used to detect NiV only. Using these two ELISAs, we can differentiate between NiV and HeV. The results showed that both NiV and HeV were detected in the ELISA using mAb F27NiV-34, whereas only NiV was detected using mAb F20NiV-65 (Figure 5a).”

·        Line 225: Ephrin B2 was selected as the capture antigen based on their previous study, and this has been mentioned in the discussion. It might still be a good idea to add the reference here.

·        Line 227: If the manuscript describes an optimization and makes a related recommendation, data should be included, or it should not be discussed at all. Optimizations on plates and mAb concentrations are described in “Development of AgELISAs,” which is fine, but without any data to support that, I don’t see any contribution of these. I recommend either removing this part and changing the title to something like “Assay cutoff determination” or including the data.

·        Figure 4: The titer of inactivated viruses used to detect the analytical specificity of the AgELISAs is not given. Also, there is not much Information available on NiV-M, NiV-B, and HeV cell culture practices. How was the titer determination performed? Are these reagents the same preparations as those in the previous study? This should be referred to, or the sentence should be edited to address this.

·        Figure 4: The first half of this figure is also in the previously published study. EBOV and PBS negative control and the effect of lysis buffer on detection are new data. These should be graphed, and the rest should be referred.

·        Line 267: In their previous study, They say both mAbs are binding to conformational epitopes.

·        Line 271: This should be Figure 5 instead of Figure 4.

·        Figure 5: Would it be better if this data is moved to “Characterization henipavirus-specific mAbs”?

·        Table 2b, HeV instead of HiV.

·        Experimental replicates should be included in all graphs when available with median/mean and SD.

·        Line 305-307: Again, maybe I am missing something here, but this has been published before; if there are some modifications compared to the previous study, what are those?

·        Line 315: “while the binding site of F27NiV-34 is located on NiV F”, this is the only data described in this entire paragraph that has been generated in this paper.

Author Response

Reviewer 1:

The authors describe an AgELISA assay to detect NiV and HeV. The importance of having an established assay and its possible uses have been described. Some of the data is similar to what has been published before by the authors. The manuscript should focus on the new information it brings.

Minor comments:

  • Line 143: This section might need a better description of the source of the samples. Three pig samples used in the study were control samples from a previously unpublished study. Using “control animals” may be a little confusing: positive or negative control?  Maybe just say banked samples of NiV-M-infected animal samples from a previous study.

Answer: The sentence was changed from "the previous study" to "stored NiV-M-infected animal samples".

Oral and nasal swabs were collected from 40 animals prior to virus inoculation, but only three of them got infected?

Answer: Yes, 80 oral and nasal swab samples were collected from 40 animals prior to virus inoculation. Three samples out of 80 tested false positive using the NiV/HeV AgELISA.

  • Line 114: The concentrations of proteins used for coating ELISA plates can be added to the method section.

Answer: The concentrations of the coating proteins have been added.

  • Table 1: Any IHC performed with these samples?

Answer: No, we didn’t perform IHC with these samples.

  • The data in Figure 1a has already been published before. What is the mAb dilution used in this assay?

Answer: The data in Figure 1a has not been published before. The mAb dilution (1:50) was added.

  • Lines 195-201 and most of Figure 2 have been published by the authors previously.

From the previous paper:

“Given that ephrin B2 can replace polyclonal antibodies as a ligand in ELISAs to detect henipavirus, we next sought to establish a more specific antigen detection ELISA using mAbs F27NiV-34 and F20NiV-65. Since mAb F27NiV-34 reacts with both NiV and HeV, it was used to detect both henipaviruses, while mAb F20NiV-65, which reacts only with NiV, was used to detect NiV only. Using these two ELISAs, we can differentiate between NiV and HeV. The results showed that both NiV and HeV were detected in the ELISA using mAb F27NiV-34, whereas only NiV was detected using mAb F20NiV-65 (Figure 5a).”

Answer: Figure 2 shows mAb reactivity against NiV-B and HeV. Yes, this information was stated in the previously published results. However, in this report, we further confirmed previous findings by testing two-fold serial dilutions of NiV and HeV. New results demonstrate that mAbs bind NiV and HeV in a dose-dependent and virus-specific manner.

  • Line 225: Ephrin B2 was selected as the capture antigen based on their previous study, and this has been mentioned in the discussion. It might still be a good idea to add the reference here.

Answer: The required reference was added.

  • Line 227: If the manuscript describes an optimization and makes a related recommendation, data should be included, or it should not be discussed at all. Optimizations on plates and mAb concentrations are described in “Development of AgELISAs,” which is fine, but without any data to support that, I don’t see any contribution of these. I recommend either removing this part and changing the title to something like “Assay cutoff determination” or including the data.

Answer: The title has been changed from “Development of AgELISAs,” to “Assay cutoff determination”.

  • Figure 4: The titer of inactivated viruses used to detect the analytical specificity of the AgELISAs is not given. Also, there is not much Information available on NiV-M, NiV-B, and HeV cell culture practices. How was the titer determination performed? Are these reagents the same preparations as those in the previous study? This should be referred to, or the sentence should be edited to address this.

Answer: The references to cell culture and viral titration procedures (NiV, HeV, recombinant VSV, and EBOV) have been added to the manuscript.

  • Figure 4: The first half of this figure is also in the previously published study. EBOV and PBS negative control and the effect of lysis buffer on detection are new data. These should be graphed, and the rest should be referred.

Answer: Yes, the first half of Figure 4 is similar to previously published data. However, the data in Figure 4 are new data after extensive optimization of the ELISA.

  • Line 267: In their previous study, They say both mAbs are binding to conformational epitopes.

Answer: The previous study was a preliminary study on mAb characterization.

In this study, the binding epitopes of mAgs were comprehensively evaluated. After extensive characterization testing, we confirmed that the binding epitope of F20NiV-65 is linear and F27NiV-34 is conformational.

  • Line 271: This should be Figure 5 instead of Figure 4.

Answer: This has been corrected.

  • Figure 5: Would it be better if this data is moved to “Characterization henipavirus-specific mAbs”?

Answer:  The data in Figure 5 are presented to further confirm that the AgELISA is NiV-specific. Thus it is more suitable for the "AgELISA Assay Specificity and Sensitivity" section.

Table 2b, HeV instead of HiV.

Answer: This has been corrected.

  • Experimental replicates should be included in all graphs when available with median/mean and SD.

Answer: Experimental replicates have been added to all graphs.

  • Line 305-307: Again, maybe I am missing something here, but this has been published before; if there are some modifications compared to the previous study, what are those?

Answer: Previous studies aimed to approve Ephrin B2 for use as a capture agent in immunoassays. In this report, we focused on mAb characterization, ELISA development, and evaluation. Since the last publication, the ELISA has been thoroughly optimized and evaluated in terms of specificity and sensitivity.

  • Line 315: “while the binding site of F27NiV-34 is located on NiV F”, this is the only data described in this entire paragraph that has been generated in this paper.

Answer:  In this paragraph, we intended to provide background and explain why these two mAbs were chosen for use in the AgELISA.

Reviewer 2 Report

Comments and Suggestions for Authors

The manuscript by Zhu et al describes the development of an ELISA for the detection of Nipah and Hendra viruses that with potential to be used for rapid detection in large numbers of samples and suitable for use in remote areas. Overall the manuscript is well written and provides a new assay and reagents to the field. Just a couple of minor comments.

Although the assay is tested against a number of different viruses with no evidence of cross reactivity, the authors should comment on other henipaviruses, including the second Hendra virus genotype identified. Is there potential for cross reactivity with the second Hendra variant and also with other henipaviruses identified in bats and other mammals?

Although the authors indicate the assay is suitable for large numbers of samples and in remote areas, the sample volumes required for ELISAs are often high. Is there potential to transfer this to the Luminex platform? Considering the assay would need to be performed with both mAbs for detection of Hendra virus, this could be a potential limiting factor in the use of the assay.

Detection of Nipah virus in experimentally infected pig samples is quite limited despite the low Ct values detected by PCR. The authors should comment on the expected identification of viral antigen in pig samples based on previous studies in the field.

Author Response

Thank the reviewers for taking the time to carefully read through our manuscript.

Although the assay is tested against a number of different viruses with no evidence of cross reactivity, the authors should comment on other henipaviruses, including the second Hendra virus genotype identified. Is there potential for cross reactivity with the second Hendra variant and also with other henipaviruses identified in bats and other mammals?

Answer: We agree. This is a very good point. However, due to limitations in obtaining second Hendra variant and other henipaviruses, we are unable to test them. If possible, we will try to test those virus to fully validate the ELISA.   

This limitation has been discussed in the manuscript.

Although the authors indicate the assay is suitable for large numbers of samples and in remote areas, the sample volumes required for ELISAs are often high. Is there potential to transfer this to the Luminex platform? Considering the assay would need to be performed with both mAbs for detection of Hendra virus, this could be a potential limiting factor in the use of the assay.

Answer: We agree. Current AgELISA formats may not be ready for transfer to the Luminex platform.

Detection of Nipah virus in experimentally infected pig samples is quite limited despite the low Ct values detected by PCR. The authors should comment on the expected identification of viral antigen in pig samples based on previous studies in the field.

Answer: This is a very good point. We addressed this issue in discussion.

Reviewer 3 Report

Comments and Suggestions for Authors

Interesting work! Some editing and additional experimentation is recommended to validate some statements. Please find below my comments:

-Rephrase line 39-40, 44-45, 48-49, 51-53, 77-84, 103-110.

-Missing biocontainment level in M&M

- cannot say specific for henipavirus in the manuscript since you have not tested against LayV, Cedar, and others.

-Check dose of irradiation

-How did you culture and titrate viruses (NiV, HeV, recombinant VSV, and EBOV)?

-rephrase 132-134. How much of antibody did you use?

-Table 1 presents results of ELISA before determining Spe, Sen, and LOD. should go after figure 5 and table 2. Data from neg samples could also be presented in table 1 so that you could calculate the sensitivity from these.

-How much of mAbs in line 190? How long is the incubation time?

-Data from naive sera would be good to see in figure 1 and 2

-line 196: unclear if inactivated or infectious

-Fig 4: add a line for cutoff

-be consistent in the color-code used for antibodies.

-Line 266 and Fig4: NiV/HeV Ag ELISA has signal above cutoff so still positive in lysates. linear vs. conformational still remains valid.

-What was the concentration of EBOV in Fig4 (related to next question)?

-Fig 5: using vsv EBOV at the same concentration than the counterpart with NiV would be helpful. Specific signal is acquired before 1:64 dilution so one could argue that 2log 10 difference in titer explain this result and that your ELISA is picking up EBOV GP (Although unlikely?) Was the titration method the same?

-LIne 271: do you mean fig5?

-Table 2: HeV not HiV.

-Line 319-320: not valid until proven by new figure 5.

-Table 3 not needed in discussion. Use one or 2 data in the text directly.

Comments on the Quality of English Language

English is fine

Author Response

We thank the reviewers for taking the time to carefully read our manuscript. Our replies to your comments and questions are as follows.

-Rephrase line 39-40, 44-45, 48-49, 51-53, 77-84, 103-110.

Answer: Those sentences have been rephrased.

-Missing biocontainment level in M&M

Answer:  Biosafety level 4 laboratories (BSL-4) has been added.

- cannot say specific for henipavirus in the manuscript since you have not tested against LayV, Cedar, and others.

Answer:  This is a very good point. Henipavirus has been replaced by NiV and HeV.

-Check dose of irradiation

Answer:  The dose of the irradiation has been checked and it is correct.

-How did you culture and titrate viruses (NiV, HeV, recombinant VSV, and EBOV)?

Answer: The references to cell culture and viral titration procedures (NiV, HeV, recombinant VSV, and EBOV) have been added to the manuscript.

-rephrase 132-134. How much of antibody did you use?

 Answer:  The sentence has been rewritten and antibody dilution added.

-Table 1 presents results of ELISA before determining Spe, Sen, and LOD. should go after figure 5 and table 2. Data from neg samples could also be presented in table 1 so that you could calculate the sensitivity from these.

Answer:  The purpose of table one is to provide animal serum samples with different testing applied. We put it in method section. The table 2 is about the sensitivity, so we put it after figure 5 which is about specificity.

-How much of mAbs in line 190? How long is the incubation time?

Answer:  This information was added to the figure legend.

-Data from naive sera would be good to see in figure 1 and 2

Answer:  Figure 1 demonstrated mAb reactivity against recombinant NiV structural proteins in an indirect ELISA. Figure 2 showed mAb reactivity against NiV-B and HeV. There was no serum involved.

-line 196: unclear if inactivated or infectious

Answer:  It is inactivated virus. This was clarified.

 -Fig 4: add a line for cutoff

Answer: A cutoff line has been added to Figure 4.

-be consistent in the color-code used for antibodies.

Answer: Changes have been made and the antibodies are now consistent in color.

-Line 266 and Fig4: NiV/HeV Ag ELISA has signal above cutoff so still positive in lysates. linear vs. conformational still remains valid.

Answer: We did not test the three false-positive samples treated with lysis buffer. We may further investigate the cause of the false-positive results.

-What was the concentration of EBOV in Fig4 (related to next question)?

 Answer: The concentration of EBOV has been added to Figure 4 legend.

-Fig 5: using vsv EBOV at the same concentration than the counterpart with NiV would be helpful. Specific signal is acquired before 1:64 dilution so one could argue that 2log 10 difference in titer explain this result and that your ELISA is picking up EBOV GP (Although unlikely?) Was the titration method the same?

Answer: We agree. Ideally, the same concentration of VSV pseudotyped virus as NiV would be used. In Figure 5, we intend to demonstrate that the NiV AgELISA can detect live pseudotyped viruses since only gamma-irradiated NiV was previously tested. NiV AgELISA detected only positive results for VSV-EBOV-GP/NiV-G but negative results for VSV-EBOV-GP.

-LIne 271: do you mean fig5?

Answer: Yes, it is Fig 5. The mistake has been corrected.

Answer: -Table 2: HeV not HiV.

Answer: The mistake was corrected.

-Line 319-320: not valid until proven by new figure 5.

Answer: The results in Figure 5 demonstrated that the NiV AgELISA is specific for NiV.

-Table 3 not needed in discussion. Use one or 2 data in the text directly.

Answer: Table was removed and the data have been added in the discussion section.

Round 2

Reviewer 3 Report

Comments and Suggestions for Authors

comments have been addressed

Comments on the Quality of English Language

Minor typo to be fixed by editing board.